# Clinical, Cognitive and Neurodevelopmental Profile in Tetrasomies and Pentasomies: A Systematic Review

**DOI:** 10.3390/children9111719

**Published:** 2022-11-09

**Authors:** Giacomina Ricciardi, Luca Cammisa, Rossella Bove, Giorgia Picchiotti, Matteo Spaziani, Andrea M. Isidori, Franca Aceti, Nicoletta Giacchetti, Maria Romani, Carla Sogos

**Affiliations:** 1Section of Child and Adolescents Neuropsychiatry, Department of Human Neuroscience, Sapienza University of Rome, 00185 Rome, Italy; 2Advanced Endocrine Diagnostics Unit, Department of Experimental Medicine, Policlinico Umberto I Hospital, Sapienza University of Rome, 00161 Rome, Italy; 3Post-Partum Disorders Unit, Department of Human Neuroscience, Sapienza University of Rome, 00185 Rome, Italy

**Keywords:** developmental disorders, neurocognitive profile, neuropsychological profile, sex chromosome aneuploidie (SCAs), sex chromosome pentasomy, sex chromosome tetrasomy

## Abstract

*Background*: Sex chromosome aneuploidies (SCAs) are a group of disorders characterised by an abnormal number of sex chromosomes. Collective prevalence rate of SCAs is estimated to be around 1 in 400–500 live births; sex chromosome trisomies (e.g., XXX, XXY, XYY) are most frequent, while tetra- and pentasomies (e.g., XXXX, XXXXX, XXXY, XXXXY) are rarer, and the most common is 48, XXYY syndrome. The presence of additional X and/or Y chromosomes is believed to cause neurodevelopmental differences, with increased risk for developmental delays, language-based learning disabilities, cognitive impairments, executive dysfunction, and behavioural and psychological disorders. *Aim of the Study*: Our review has the purpose of analysing the neurocognitive, linguistical and behavioural profile of patients affected by sex chromosomes supernumerary aneuploidies (tetrasomy and pentasomy) to better understand the specific areas of weakness, in order to provide specific rehabilitation therapy. *Methods*: The literature search was performed by two authors independently. We used MEDLINE, PubMed, and PsycINFO search engines to identify sources of interest, without year or language restrictions. At the end of an accurate selection, 16 articles fulfilled the inclusion and exclusion criteria. *Results and Conclusions*: International literature has described single aspects of the neuropsychological profile of 48, XXYY and 49, XXXXY patients. In 48, XXYY patients, various degrees of psychosocial/executive functioning issues have been reported and there is an increased frequency of behavioural problems in childhood. Developmental delay and behavioural problems are the most common presenting problems, even if anxiety, depression and oppositional defiant disorder are also reported. They also show generalized difficulties with socialization and communication. Cognitive abilities are lower in measures of verbal IQ than in measures of performance IQ. Visuospatial skills are a relative strength compared to verbal skills. In patients with 49, XXXXY, both intellectual and adaptive functioning skills fall into the disability range, with better non-verbal cognitive performance. Speech and language testing reveals more deficits in expressive language than receptive language and comprehension. Anxiety, thought problems, internalizing and externalizing problems, and deficits in social cognition and communication are reported. Behavioural symptoms lessen from school age to adolescence, with the exception of thought problems and anxiety. Individuals affected by sex chromosome aneuploidies show testosterone deficiency, microorchidism, lack of pubertal progression and infertility. Hormone replacement therapy (HRT) is usually recommended for these patients: different studies have found that testosterone-based HRT benefit a wide range of areas initiated in these disorders, affecting not only neuromotor, cognitive and behavioural profile but also structural anomalies of the brain (i.e., increase of volume of grey temporal lobe matter). In conclusion, further studies are needed to better understand the neuropsychological profile with a complete evaluation, including neurocognitive and psychosocial aspects and to establish the real impact of HRT on improving the cognitive and behavioural profile of these patients.

## 1. Introduction

Sex chromosome aneuploidies (SCAs) are a group of disorders that are characterised by an abnormal number of X or Y chromosomes. SCAs are the most commonly occurring aneuploidies in children with a collective prevalence rate of about 1 in 400–500 live births [1,2]. They include conditions in which there is either a missing or an extra sex chromosome (s). The most common SCAs are trisomies (XXX in females, XXY and XYY in males), followed by monosomy X in females (Turner syndrome) [3].

47, XXY (Klinefelter syndrome—KS) is the most common X chromosome aneuploidy, and its incidence rate is approximately 1 in 750 male births [4], for which there is rich literature delineating the physical and neuro behavioural phenotype. Among the rarer X chromosome aneuploidies, known as high-grade aneuploidies (HGAs), of sexual chromosomes with male phenotype, 48, XXYY syndrome is the most common (1 in 18,000–50,000 male births). The incidence of 48, XXXY is among 1–9 in 100,000 male births; 49, XXXXY is estimated to be 1 in 85,000–100,000 male births, and 49, XXXYY is the rarest form [5]. Even if the more complex chromosome aneuploidies 48, XXYY, 48, XXXY, and 49, XXXXY are often associated with 47, XXY, due to shared features, there is a wider spectrum of physical and cognitive abilities that underlies distinctive features between these conditions [6]. A 2018 paper first demonstrated that there are endocrine and metabolic differences between classic KS and HGAs, so that these conditions can be categorised as two different and distinct conditions [7].

The addition of extra X and/or Y chromosomes leads to neurodevelopmental differences, with an increased risk of developmental delays, language-based learning disabilities, cognitive impairment, executive dysfunction and behavioural and psychological disorders [8]. Data in the literature have shown that individuals with SCAs have an increased risk of impairment in verbal fluency tasks (semantic and phonemic fluency), and in cognitive domains, including episodic memory, processing speed, and different executive functions [9]. There is evidence that this impairment in tetrasomies and pentasomies is associated with supernumerary chromosomes: an increase in the number of X chromosomes is associated with increased verbal fluency impairments [9]. The nature and severity of the impairment depends on the patient group: specifically, an extra Y chromosome was associated with overall performance decrement, while an extra X chromosome was more associated with phonemic than semantic fluency impairment [9]. A study conducted by Gropman and Samango-Sprouse [6] demonstrated that, despite less impairment in the visuo-spatial domain compared to the verbal domain, X and Y polysomies were also associated with moderate-to-severe developmental dyspraxia. Decreased motor control and motor abilities are evident in males with 47, XXY and 49, XXXXY [6]. Due to dyspraxia in oral motor and verbal abilities and motor planning deficits, most 47, XXY boys have a lag in early expressive language skills with delayed acquisition of single words and phrases [6]. Language formulation is more affected than receptive skills. The severity of language-based learning deficits in this group varies from moderate to severe and affects their ability to develop social interactions, resulting in behavioural manifestations of frustration and oppositional behaviour [6]. The deleterious effects on physical and cognitive development increase with the number of supernumerary X chromosomes [10]. These findings provide further support the possible differential effects of supernumerary X and Y chromosomes on verbal fluency performance, whereas other studies are required to better define the different clinical profiles associated with different SCAs [9]. Our study aimed to analyse the neurocognitive, linguistic and behavioural profile of patients affected by supernumerary SCAs, specifically tetrasomy and pentasomy. We investigated the verbal abilities, both expressive and receptive, as well as the metalinguistic comprehension and attentive skills of these patients.

## 2. Materials and Methods

### 2.1. Review Protocol

The protocol of this systematic review is available on the PROSPERO website at the link https://www.crd.york.ac.uk/prospero/ (accessed on 28 August 2022) and whose registration number is CRD42022354270. The Preferred Reporting Items for Systematic Reviews and Meta-Analyses (PRISMA) guidelines were used as a template when performing the Review.

### 2.2. Literature Search

We used MEDLINE, PubMed and PsycINFO search engines to identify sources of interest, without year or language restrictions. Search terms were: ((SCAs) OR (sex chromosome polysomy) OR (sex chromosome pentasomy) OR (sex chromosome tetrasomy) OR (Sex chromosome tetrasomy andpentasomy) OR (49XXXXY) OR (48XXYY) OR (48XXXY) OR (48XXXX) OR (49XXXXX) OR (49XXXYY) OR (48XYYY) OR (49XYYYY) OR (49XXYYY)) AND ((child) OR (children)) AND ((development) OR (neurocognitive profile) OR (cognitive)). The search has been performed by two authors independently. All references of the selected articles were searched for relevant studies not identified by electronic searches. 

To try and define temporal restriction, we searched for the last published review focused on the neuropsychological and cognitive profile in sexual pentasomy and tetrasomy, but found only 1 review published in 1995 that was not available in full text. Thus, we proceeded to analyze all the found articles. After thoroughly reviewing these studies, we defined inclusion and exclusion criteria.

### 2.3. Inclusion and Exclusion Criteria

Studies were included in this review if they focused on neuropsychological and/or cognitive profile, emotional-behavioural issues, neurodevelopment, pentasomy and tetrasomy, and includes 3 or more patients in the study who were aged between 0–25. We included articles on pharmacological strategies in order to relate these strategies with the outcome. Exclusion criteria were a focus on prenatal screening or parental point of view, neuroimaging, or other aneuploidies.

### 2.4. Quality Assessment

Quality assessment of the included literature was determined using the National Institutes of Health (NIH) Quality Assessment Tool. Table 1 shows results of this evaluation, performed by two independent reviewers.

## 3. Results

### Search Results

These searches returned 574 articles. We then eliminated duplicates and were left with 487 studies. We then reviewed titles and abstracts and eliminated all articles that did not focus on pentasomy and tetrasomy. This left us with 121 studies. We then excluded articles that were not in English or Italian, which left 115 studies. At this point, two independent authors selected those articles that focused on neuropsychological and/or cognitive profile, emotional-behavioural issues, neurodevelopment, pentasomy, and tetrasomy, and included three or more patients in the study who were aged between 0–25. After our selection, 16 articles fulfilled the inclusion criteria.

Figure 1 shows the detailed selection process.

Table 2 shows a summary of the articles selected, reporting locations of recruitment, participants number, age of participants, assessments used and lastly the main results of the studies.

## 4. Discussion

### 4.1. 48, XXYY SYNDROME

48, XXYY syndrome was described for the first time by by Muldal and Ockey, in 1960, in a 15-year-old boy with clinical features associated with KS. Males with 48, XXYY tend to have small testes, tall stature, low levels of testosterone and infertility [4,22]. The phenotypes of KS and 48, XXYY syndromes overlap, even if the latter syndrome is probably associated with more medical problems and more severe neurodevelopmental and behavioural difficulties [23,24]. In addition, intellectual impairment seems to be more severe (borderline or intellectual disability) in males with 48, XXYY than in those with KS [22,25,26]. As already stated, this condition occurs in 1 in 18,000–40,000 males [10,27].

#### 4.1.1. Neurological Features

The neurological phenotype of the first 53 described cases of 48, XXYY included seizures, intentional tremor, hypotonia and tics [6]. Other studies also showed postural and kinetic tremors that often worsened over time associated with five coordination disturbances [28,29].

#### 4.1.2. Medical Aspects

These patients show an increased risk of reactive airway disease, asthma, allergies, obstructive sleep apnoea and seizure disorders [24,26,30]. The physical phenotype includes cleft palate, poor dentition, strabismus scongenital hip dysplasia and heart defects, pes planus, clubfoot, scoliosis and renal dysplasia [24,26,30]. Other medical issues reported include deep vein thrombosis, pulmonary embolism, constipation, recurrent otitis media, gastroesophageal reflux disease, hypothyroidism, type 2 diabetes mellitus and an increased mortality due to non-Hodgkin’s lymphoma [6,8,24,26,31]. Typical facial dysmorphisms are hypertelorism (epicanthal folds and narrow upslanting palpebral fissures) [16]. 48, XXYY individuals have testicular hyalinization, which leads to testosterone deficiency and microorchidism, with a lack of pubertal progression and infertility [22]. For this reason, testosterone replacement therapy is usually recommended for this group of patients [16,26].

#### 4.1.3. Psychosocial and Behavioural Aspects

Various degrees of psychosocial/executive functioning issues have been reported by Muldal and Ockey in 1960 [29,32]; Barr et al., Hunter, and Casey described the association with increased behavioural problems and aggressive offences, such as those against both inanimate objects and individuals, in childhood [29]. These individuals can have difficulties with socialization and communication but can also present strengths in adaptive daily living skills [15]. Anxiety and depression are common [8,24]. Oppositional defiant disorder (ODD) has been reported with a range of frequencies (from 10 to 46%) [24]. An increased risk of psychosis in adult life is also reported [4,23,24,28,33,34,35]. Other emotional symptoms (i.e., emotional immaturity, obsessive-compulsive behaviours, impulsivity, behavioural dysregulation, and tic disorders) are more commonly seen in 48, XXYY than in 47, XXY patients or the general population [24].

Psychosocial features in 48, XXYY patients are reported in a 1978 study by Sørensen et al. [27]: a tendency towards confabulation of stories, preferential socialization with younger people, introversion, episodic violent or criminal acts are described in their work. Afterwards, several studies, including works by Tartaglia et al. and Visootsak et al. [14,36], have shown analogous features: findings include meticulous descriptions of these traits, a higher amount of dysfunctional behaviours, internalizing and externalizing behaviours, autism spectrum disorder (ASD) or ASD-like symptoms, and attention-deficit hyperactivity disorder (ADHD) in comparison with 47, XXY patients [9]. Other studies describe that 48, XXYY patients show better outcome in socialization, communication and daily living skills in comparison with individuals affected by other SCA tetrasomies and pentasomies (Tartaglia et al.) [8,24,26]. It was reported that psychopharmacological treatment was received by about 50% of these patients, mainly to handle attention problems, anxiety, impulsivity, and mood instability; hospital care was necessary in one third of this population [24,26]. Numerous studies in the late 2000s by Tartaglia et al. [8,14] and Cordeiro et al. [12] have shown findings different from those described in the past: in particular, they reported that psychosocial issues were definitely not as prevalent or severe as previously described. In the last 10–15 years, studies about 48, XXYY patients have described specific psychosocial findings that are distinctive for these individuals, as well as positive and valuable affective and behavioural features that are usually present [29]. A more recent study by Srinivasan et al. [16] indicates that temper outbursts are specifically associated with the 48, XXYY phenotype, though difficulties with social communication and attention/concentration may be associated with intellectual impairment. A higher risk to develop academic, behavioural, and social deficits is reported in 48, XXYY patients [25,30]. In a 2008 study by Tartaglia et al., language-based tasks show worse outcomes than those evaluating visuoperceptual skills in 48, XXYY patients. Furthermore, significant deficits in adaptive functioning scores (especially in areas related to communication, social skills, self-care, and self-direction) were reported [26]. In these individuals, processing/integration disorders, tic disorders, speech apraxia, and auditory processing disorders were also noticed [7,29]. Psychopharmacologic medications to manage behavioural and psychiatric symptoms were effective in sex chromosome tetrasomy and pentasomy patients and are important to consider in conjunction with behavioural therapies for moderate-to-severe behavioural or emotional symptoms [24,29].

#### 4.1.4. Neurodevelopmental Aspects

Patients with 48, XXYY present delays in developmental milestones (both speech and motor delays) [29,37]. Over 90% of children present with speech and language delays and over 75% have motor development delay in early childhood [14,16,26]. Neurodevelopmental disorders are very common: ADHD is found in nearly three out of four patients, ASD or ASD-like behaviours are found in up to half of cases [4,8,24,25,26,29]. Many studies report difficulties with social skills [12,24,36]. Cordeiro et al. [12] found that 37.5% of children with 48, XXYY had a prior diagnosis of ASD, and 43.8% had severe scores on a measure of autistic traits (the social responsiveness scale (SRS)). Furthermore, even patients without a diagnosis of ASD showed a behavioural phenotype that overlapped with behaviours seen in ASD [14,38,39]. Many of these overlapping behaviours, including language delays, poor eye contact, oversensitivity to sensory stimuli, poor social relatedness, and restricted interests, have been previously reported in children with SCA [14,38,39]. In a study conducted by Tartaglia et al. [14] in 2017, it was noted that males with Y chromosome aneuploidies (they were 47, XYY and 48, XXYY) had an increased prevalence of ASD and more severe autistic features in those who did not meet the diagnostic criteria for ASD as compared to males with KS and a normal karyotype, due to a probable role of the Y chromosome in determining ASD susceptibility [40]. ADHD symptoms are very common in all SCA groups, present in 3 to 10%, with the inattentive subtype being the most common [8]. However, males with 47, XYY and 48, XXYY are more likely to present associated symptoms of hyperactivity and impulsivity [8]. Attentional difficulties may be due to other overlapping conditions, such as hypothyroidism, seizure disorders, sleep apnoea and testosterone deficiency [8]. Thus, evaluation of testosterone levels must be considered and treatment optimized [8]. More than 70% of children and adolescents with SCA and ADHD respond to standard stimulant medications, with a relatively low rate of significant side effects, with the most common being increased irritability, which may lead to treatment discontinuation [8]. For this reason, low starting doses with gradual increases are recommended to decrease the impact of irritable symptoms [8].

#### 4.1.5. Cognitive Phenotype

Cognitive abilities are lower in measures of verbal IQ than in measures of performance IQ. Visuospatial skills show relative strength compared to verbal skills. In general, the greater the number of supernumerary sex chromosomes, the greater the impact of the aneuploidy on intelligence [41,42]. IQ scores are in the borderline-to-intellectually disabled range [4,22,23,24,25], with the majority having an IQ in the 70–80 range [29,43]. However, in previous studies, all SCAs were considered intellectually disabled, and studies by Tartaglia et al. [6] and Gropman and Samango-Sprouse [24] have shown a larger spectrum of cognitive abilities in these patients [29]. More recent articles also indicate that verbal IQ is lower in adults than in children, whereas there are no significant differences in visuo-spatial IQ [16,24]. It is interesting that some boys who received early treatment and services had an IQ in the low normal range [29]. Cognitive abilities did not predict adaptive functioning [29]. Visootsak et al. [25] compared the results of adaptive functioning using Vineland Adaptive Behaviour Scales in a cohort of patients with tetrasomies and pentasomies. The mean scores were in the disability range. Boys with 48, XXYY showed significantly higher scores in the domains of daily living skills and communication compared to boys with 48, XXXY or 49, XXXXY [25]. They found no significant differences in social skills [6,25]. In 2017, Printzlau et al. [44] compared cognitive abilities in children with 48, XXYY, 47, XXY/XYY and 49, XXXYY/XXXXY, and found that the first group had lower abilities than the second one, but higher abilities than the third [16]. In a cohort of patients with 48, XXYY, adaptive functioning was found to be significantly lower than IQ in most cases, with a mean score of 68.9 [6]. These findings suggest that daily functioning is often more impaired than would be expected for these patients based on cognitive (IQ) scores alone. The reason for these deficits in adaptive skills could be due to motor planning deficits and hypotonia [6].

#### 4.1.6. Structural Neuroimaging Studies

Structural imaging findings in 48, XXYY syndrome show that the volume of the brain is reduced by about 5% [45]. Evidence from other SCAs suggests that total brain volume decreases with each additional X chromosome [41]. Differently, in the 2015 study by Hanley et al. [45], brain abnormalities were attributed to the total number of supernumerary sex chromosomes rather than to the impact of an additional X or Y chromosome specifically. There were areas of white matter hyperintensity as well as enlarged ventricles in a large minority of cases [6,23,41,45]. There was a disproportionate excess of both grey matter and white matter in the parietal lobe. This could imply a potential dissociation of the influence of additional X and Y chromosomes on the temporal and parietal lobes, respectively. Other abnormal, but non-specific findings in some boys included agenesis of the corpus callosum or frontoparietal cortical atrophy, corpus callosum lipoma, cortical dysplasia, and, rarely, pituitary adenoma [6,29,41]. In addition, Wade et al. [46] suggested a potential role of X- and Y-linked genes in corpus callosum morphometry. In 2016, Reardon et al. [47] highlighted a reduction in pallidal volume of the basal ganglia in individuals with SCAs; they did not find significant differences in cortical asymmetry and cortical folding [29,48,49].

### 4.2. 49, XXXXY SYNDROME

Most sex chromosome variations are the result of the addition of a single X or Y chromosome, but according to historical data, three extra X chromosomes can occur in around 1 of 85,000 to 100,000 live male births [23,50]. 49, XXXXY syndrome is a rare disorder that is due to double non-disjunction of the replicating X chromosome during both meiosis I and II, and it was firstly reported in 1960 [8]. Patients affected can be diagnosed prenatally (via amniocentesis and chorionic villus sampling) or with postnatal diagnostic testing. They show specific dysmorphic features, organ anomalies, severe mental retardation and congenital malformation, which can lead to a diagnosis in the first year of life. [18]. Some authors hypothesised that the overall features of affected patients were due to the abnormal dosage of genes beyond what is normally guaranteed between males and females by X-inactivation of the second X chromosome in females [21]. It is known that clinical, neurocognitive, and developmental characteristics worsen with extra X chromosomes [37]. Gropman and colleagues [10] studied clinical variability and neurodevelopmental findings in 20 patients affected by 49, XXXXY and attributed the variability of their profile to skewed inactivation of additional chromosomes or mosaicism and methylation abnormalities [21]. According to Lee et al. [21], methylation levels of multiple CpGs in androgen receptor (AR) monoamine oxidase A (MAO-A) genes are associated with differences in both internalizing and externalizing behaviour in boys with 49, XXXXY as compared with neurotypical boys.

#### 4.2.1. Clinical Features and Neurological Examination

Common physical and clinical features of 49, XXXXY subjects include facial dysmorphisms (upslanting palpebral fissures in 100 % of cases, arched eyebrows, ocular hypertelorism, synophrys, flat nasal bridge, frontal bossing, low-set ears), cardiac defects, endocrinologic abnormalities (e.g., hypogonadism, hypogenitalism, growth retardation), microcephaly, short stature, multiple skeletal anomalies, mental retardation and speech delay [10,15,18]. Other features were less frequently observed and consisted of alterations in muscular tone (generalised hypotonia), joint laxity, cleft palate, bifid uvula, kidney and ear malformations, and other skeletal anomalies (such as plagiocephaly, pectus excavatum, pes cavus, radioulnar synostosis, fifth finger clinodactyly, syndactyly and genu valgum) [10,15,18]. In 2019, Burgemeister et al. [15] focused on dental anomalies that can occur throughout life, such as delayed dentition and eruption, hypodontia and malocclusion [18,50]. Some features are similar to those found in Down syndrome patients, and as a result, 49 aneuploidies were sometimes previously diagnosed after chromosomal testing to evaluate the possibility of trisomy 21 [18]. In the past, 49, XXXXY was associated with prenatal growth deficiency, though more recent publications show borderline growth parameters [10]. Unlike males with KS or other sexual chromosome variations, 49, XXXXY patients often have decreased stature [13]. In 2018, Burgemeister et al. [48] found that most patients outgrew perinatal growth deficiencies, including low head circumference. No cranial nerve abnormalities have been reported [18], even if some cases of ocular motor apraxia, head titubation, febrile or afebrile seizures have been described [10,15,18]. One patient described in 2018 had epilepsy due to the coexistence of Dandy–Walker malformation [15]. In most studies, electroencephalography (EEG) has not shown abnormalities [10], but in a cohort of 8 patients described by Burgemeister et al. [15] in 2018, one patient showed intermittent EEG abnormalities without clinical manifestations. In the same work, recurrent respiratory infections were reported, and one patient displayed intentional tremor [15].

#### 4.2.2. Cognitive Profile

Previous studies on aneuploidies described moderate-to-severe mental retardation in this cohort of patients, with Intelligent Quotient (IQ scores) in the range of 20–60. However, more recent publications have reported a more optimistic and less significant cognitive delay, similar to 47, XXY (KS) patients [7]. Both intellectual and adaptive functioning skills fall into the disability range [10], though Gropman et al. [10], utilising the Leiter International Performance Scale, Revised (LIPS-R), showed that nonverbal intelligence was relatively preserved compared to language skills (average non-verbal IQ of 81) [13]. IQ seems to decrease 15 points per additional chromosome, and the average IQ score is probably between 60 and 80 [10,13,15]. Another study reported a mean IQ score of 56 (range: 40 to 71) [15].

The cognitive profile may be related to white matter abnormalities, though this aspect needs to be further investigated [7].

#### 4.2.3. Behavioural Profile

Contrasting personality features characterise these patients, described as shy, friendly, oppositional, irritable, and prone to temper tantrums [10,15,51]. Furthermore, these patients do not seem to tolerate frustrations or accept changes or transition well [10,15,51]. Some studies link this behavioural profile and possible social interaction difficulties with language deficits [10,52]. It is known that 49, XXXXY patients are more affected by anxiety, thought problems, and internalizing and externalizing problems, as demonstrated by more elevated scores in different domains of the Child Behaviour Checklist (CBCL). In 2020, Lasutschinkow et al. [19] evaluated a cohort of patients with 49, XXXXY by administering the CBCL, Behaviour Rating Inventory of Executive Function (BRIEF) and SRS to parents. This study found social cognition and communication deficits beginning in preschool, though these boys presented with consistent social awareness and motivation for social activities, which were not previously appreciated in this disorder. In addition, signs of anxiety presented during preschool years and increased in severity with age, particularly in internalizing problems [19]. Behavioural symptoms lessened from school age to adolescence, with the exception of thought problems and anxiety. In particular, depressive behaviours seemed to affect less than 20% of adolescent boys with 49, XXXXY. The largest deficit seen in adolescents was in inhibiting behaviours, in which over 80% of boys had elevated scores. In addition, over 42% of adolescents had clinically elevated working memory [19]. Lee et al. [21] conducted a pilot study to understand both the association between methylation on X chromosome genes and behavioural aspects in 47, XXY and 49, XXXXY patients and the possible effect of hormonal replacement therapy (HRT). Parents of patients completed the CBCL and BRIEF questionnaires, and experts did not observe significant differences in scores on either questionnaire between 46, XY and 47, XXY groups, though higher scores related to executive dysfunction, internalizing, externalizing, and total behavioural problems were observed in the 49, XXXXY group [21]. The three extra X chromosomes in this cohort of patients were associated with important effects on gene expression and behaviour, as compared to patients with just one extra X chromosome (47, XXY) [21]. After DNA methylation analysis of a small selective group of X chromosome genes (conducted on a saliva sample), the authors concluded that methylation levels of multiple CpGs in MAO A were associated with externalizing behaviour problems, while in AR they were associated with Brief Regulation Index score [21]. In 2018, Burgemeister et al. [15] reported the case of a patient with autistic features. In the same article, persistent sleep problems, particularly concerning sleep maintenance throughout the night, were described [15]. Recent research by Gropman et al. [17] and colleagues better defined the presence of pervasive anxiety symptoms in 49, XXXXY patients.

#### 4.2.4. Neurodevelopmental Profile

Delayed neuromotor and language skills have been reported in 49, XXXXY patients, which are consistent with a decreased volume in Broca’s area and Werdmann regions, which are respectively associated with oral and verbal motor functions [10,13]. According to some authors, 49, XXXXY patients are generally able to sit between 8 and 24 months, stand between 10 and 30 months, and ambulate autonomously between 16 and 54 months [10,13]. Another study reported that the mean age of autonomous ambulation was 27 months (range: 22–36 months) [15]. These patients can also present asymmetry due to shortened pelvic and upper trunk musculature [10]. Speech and language testing reveal more deficits in expressive language than in receptive language and comprehension, configuring a pattern of developmental dyspraxia [10,15,20]. In 2021, Samango-Sprouse et al. [20] documented that childhood apraxia of speech (CAS) and associated oral motor planning deficits were pervasive in boys with 49, XXXXY, which could explain the speech and language discrepancies and deficiencies reported in this population. Furthermore, these authors compared language abilities between patients who received HRT and those who were untreated. In preschool boys, some significant differences were demonstrated between the two groups, while no significant differences between treatment groups were found in school-age children, though the treated group demonstrated less discrepancies between expressive and receptive language [20]. In particular, oral and verbal dyspraxia have been described due to the decreased tone and movements of facial musculature [10]. Non-verbal capabilities are more preserved, as demonstrated by adequate performance on non-verbal testing [10,15]. Gait is characterised by increased extension tone and reduced truncal rotation and flexion, with shortened stride. Less severe fine motor alteration, such as graphomotor dysfunctions, have also been reported [10]. Visual perception skills are also impaired in these patients [50]. More recent studies reported the increased incidence of CAS, together with motor planning deficiencies, as a possible underlying factor of expressive language delay [18].

#### 4.2.5. Neuroimaging Features

No pathognomonic MRI findings of this syndrome have been reported, though morphological, volumetric and white matter abnormalities have been described and even associated with deficits in neurodevelopmental performance [11]. Head size of these patients seems to be smaller than normal, and is associated with smaller brain volume (20% smaller than normal controls according to a study by Blumenthal et al.) [13]. Cerebral atrophy, prominent ventricles, a thinning of the corpus callosum and multiple (from 7 to more than 50 in the work by Blumenthal et al. [13]) isolated, or confluent T2-hyperintensity lesions involving periventricular and/or subcortical white matter (predominantly frontal) can be defined as the prevalent characterizing alterations in 49, XXXXY patients [13,15]. Cerebellar atrophy, cerebral hemisphere asymmetry, mega cisterna magna, arachnoid cysts and mild colpocephaly (predominantly on the left side) have also been reported, even if the latter is also described in controls [13]. Some studies have correlated a general reduction in corpus callosum size, especially in regions 3 and 4, with motor delay, dyspraxia and tactile hypersensitivity, due to the involvement of the corpus callosum in motor and sensory functions [13].

#### 4.2.6. Therapeutic Approaches

Due to their phenotypic variability and multisystemic involvement, patients with SCAs need multidisciplinary evaluation and healthcare monitoring (including from endocrinologists, neurologists, geneticists and neurodevelopmental and speech specialists). Since these patients have expressive language difficulties, with consequent irritability and frustration, they could benefit from alternative communication strategies and early targeted therapy, focused on reducing their language difficulties, and thus minimizing behavioural activation [10]. Intensive speech, physical, and occupational therapy are recommended. Different studies highlight the benefit of testosterone- based HRT, on the neuromotor, cognitive and behavioural profile of patients with 49, XXXXY, because it affects different structures of brain development (e.g., limbic areas and cortex) [11].

Testosterone treatment includes different strategies, based on the age of patient: early hormonal treatment (EHT), from 4 to 60 months of age; hormonal booster treatment (HBT), from 5 to 8 years of age and finally testosterone replacement therapy (TRT), from puberty to the rest of life [21]. Patients with high grade aneuploidies that underwent an early treatment with testosterone show persistent suppression of testicular secretory function [53]. In 2020, Gropman et al. [17], demonstrated that affected patients treated with HRT obtained higher scores on the Bayley Scales of Infant Development (BSID-III), compared with untreated patients. In the same study, testosterone treatment appeared to have beneficial effects on both verbal and non-verbal domains, throughout infancy, preschool, and early childhood [17]. In 2011, Samango-Sprouse [1] subjected 22 patients to multidisciplinary assessment (neurologic, neurocognitive, speech and language, endocrinologic), after dividing them into two equal groups: a treated (for three months) and untreated group. The authors observed that the treated group greatly improved in terms of receptive and expressive vocabulary development, gestural communication, and overall language ability. They also observed MRI changes in these patients that involved increased volume of grey temporal lobe matter (which is involved in language skills) [11].

## 5. Conclusions

International literature has described single aspects of the neuropsychological profile of 48, XXYY and 49, XXXXY patients. The aim of this review was to provide a comprehensive view of the neurocognitive, behavioural, and social aspects of these syndromes.

In 48, XXYY, various degrees of psychosocial/executive functioning issues have been reported, and there is a greater frequency of behavioural problems in childhood, with the possibility of increased aggressive offenses. Developmental delay and behavioural problems are the most common presenting problems.

Anxiety and depression are common, and oppositional defiant disorder (ODD) is also reported. Furthermore, there is an increased risk of psychosis in adult life. Other emotional symptoms reported include emotional immaturity, obsessive-compulsive behaviours, impulsivity and behavioural dysregulation. ASD or ASD-like symptoms and ADHD have been described [54,55]. Patients with 48, XXYY also have generalised difficulties with socialization and communication. However, while most children and adolescents with SCA have an interest in and are motivated by social interactions, their social cognition and social communication deficits, and autistic mannerisms are stronger contributors to their overall social difficulties.

Cognitive abilities are lower on measures of verbal IQ than on measures of performance IQ. Visuospatial skills are a relative strength compared to verbal skills. Individuals with 48, XXYY are at risk of academic deficits.

In patients with 49, XXXXY, both intellectual and adaptive functioning skills fall into the disability range, with better non-verbal cognitive performance. Delayed neuromotor and language skills are reported in 49, XXXXY patients. Speech and language testing reveals more deficits in expressive language than receptive language and comprehension, configuring a pattern of developmental dyspraxia.

Anxiety, thought problems, internalizing and externalizing problems, and deficits in social cognition and communication are reported. These patients are described as shy but friendly, though they may also appear oppositional and irritable and prone to temper tantrums. Furthermore, they may not tolerate frustrations or accept changes or transitions well. However, these patients present with consistent social awareness and are motivated to engage in social activities. Behavioural symptoms lessen from school age to adolescence, with the exception of thought problems and anxiety.

In conclusion, further studies are needed to better understand the neuropsychological profile with a complete evaluation including neurocognitive and psychosocial aspects. In addition, further studies are essential to establish the real impact of HRT on improving the cognitive and behavioural profile of these patients.

## Figures and Tables

**Figure 1 children-09-01719-f001:**
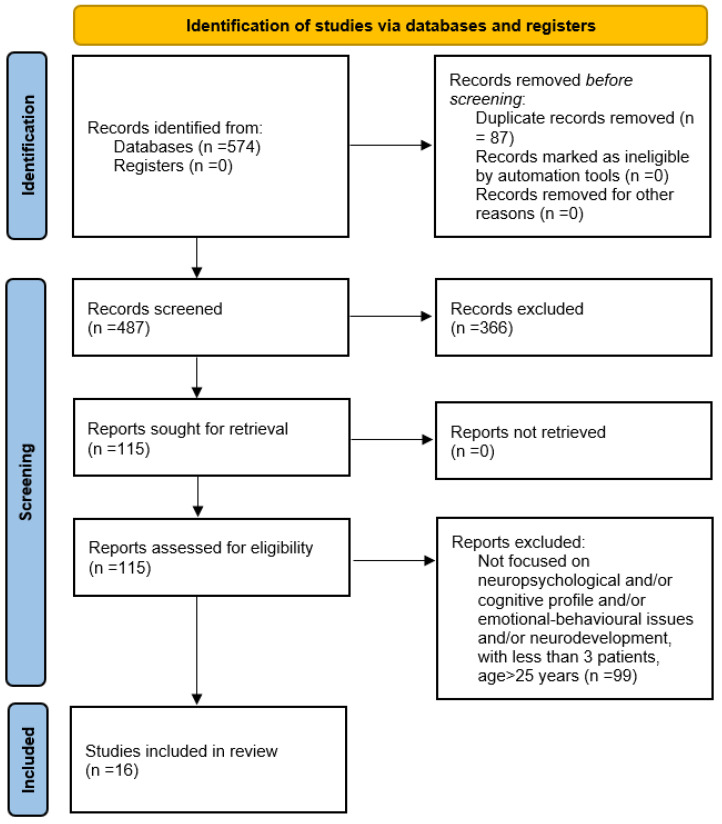
Selection process.

**Table 1 children-09-01719-t001:** Quality assessment.

**Article**	**1. Was the Study Question or Objective Clearly Stated?**	**2. Was the Study Population Clearly and Fully Described, Including a Case Definition?**	**3. Were the Cases Consecutive?**	**4. Were the Subjects Comparable?**	**5. Was the Intervention Clearly Described?**	
**Gropman et al., 2010 [10]**	YES	YES	YES	YES	YES	
**Samango-Sprouse et al., 2011 [11]**	YES	YES	YES	YES	YES	
**Lee et al., 2012 [2]**	YES	YES	YES	YES	YES	
**Cordeiro et al., 2012 [12]**	YES	YES	YES	YES	YES	
**Tartaglia et al., 2012 [8]**	YES	YES	YES	YES	YES	
**Blumenthal et al., 2013 [13]**	YES	YES	YES	YES	YES	
**Tartaglia et al., 2017 [14]**	YES	YES	YES	YES	YES	
**Udhnani et al., 2018 [9]**	YES	YES	YES	YES	YES	
**Burgemeister et al., 2018 [15]**	YES	YES	YES	YES	YES	
**Srinivasan et al., 2019 [16]**	YES	YES	YES	YES	YES	
**Gropman et al., 2020 [17]**	YES	YES	YES	YES	YES	
**Samango-Sprouse et al., 2020 [18]**	YES	YES	YES	YES	YES	
**Lasutschinkow et al., 2020 [19]**	YES	YES	YES	YES	YES	
**Thompson et al., 2020**	YES	YES	YES	YES	YES	
**Samango-Sprouse et al., 2021 [20]**	YES	YES	YES	YES	YES	
**Lee et al., 2021 [21]**	YES	NO	YES	YES	YES	
**Article**	**6. Were the Outcome Measures Clearly Defined, Valid, Reliable, and Implemented Consistently across All Study Participants?**	**7. Was the Length of Follow-Up Adequate?**	**8. Were the Statistical Methods Well-Described?**	**9. Were the Results Well-Described?**	**Reviewer 1**	**Reviewer 2**
**Gropman et al., 2010 [10]**	YES	YES	YES	YES	GOOD	GOOD
**Samango-Sprouse et al., 2011 [11]**	YES	YES	YES	YES	GOOD	GOOD
**Lee et al., 2012 [2]**	YES	NO	YES	YES	FAIR	FAIR
**Cordeiro et al., 2012 [12]**	YES	NO	YES	YES	FAIR	FAIR
**Tartaglia et al., 2012 [8]**	YES	YES	YES	YES	GOOD	GOOD
**Blumenthal et al., 2013 [13]**	YES	NO	YES	YES	FAIR	FAIR
**Tartaglia et al., 2017 [14]**	YES	NO	YES	YES	FAIR	FAIR
**Udhnani et al., 2018 [9]**	YES	NO	YES	YES	FAIR	FAIR
**Burgemeister et al., 2018 [15]**	YES	YES	YES	YES	GOOD	GOOD
**Srinivasan et al., 2019 [16]**	YES	YES	YES	YES	GOOD	GOOD
**Gropman et al., 2020 [17]**	YES	NO	YES	YES		
**Samango-Sprouse et al., 2020 [18]**	YES	YES	YES	YES	GOOD	GOOD
**Lasutschinkow et al., 2020 [19]**	YES	YES	YES	YES	GOOD	GOOD
**Thompson et al., 2020 [1]**	YES	NO	YES	YES		
**Samango-Sprouse et al., 2021 [20]**	YES	YES	YES	YES	GOOD	GOOD
**Lee et al., 2021 [21]**	YES	NO	YES	YES	FAIR	FAIR

**Table 2 children-09-01719-t002:** Articles selected.

Study	Recruitment	N	Age	Methods	Results
**Gropman et al., 2010 [10]**	Patients evaluated during consecutive summers (July 2004, 2005, 2006 and 2007) at the Neurodevelopmental Diagnostic Center for Young Children (NDCYC) in Davidsonville, Maryland.	20 boys with 49, XXXXY	Age 11 months–7 years	Leiter International Performance Scale-Revised (LIPS-R), Bayley Scales of Infant and Toddler Development, 3rd/Edition, Preschool Language Scale-3 or 4 (PLS-3/4), Peabody Motor Scale (GM, VM, total), Beery–Buktenica Developmental Test of Visual-Motor Integration, Fifth Edition (VMI), Gilliam Autism Rating Scale (GARS-2), Receptive One Word Picture Vocabulary Test-, Revised (ROWPVT-R), Expressive One Word Picture Vocabulary Test, Revised (EOWPVT-R), Dunn’s Sensory Profile for Infants and Toddlers Caregiver Questionnaire and The Sensory Profile Care- giver Questionnaire for Children (3–10 years).	Severe mental retardation (MR) in addition to craniofacial, genital, endocrine, and heart abnormalities. Previously unappreciated intact nonverbal skills are evident in conjunction with moderate to severe developmental dyspraxia.
**Samango-Sprouse et al., 2011 [11]**	A multidisciplinary clinic was held during six consecutive years (July 2004–2009) at the Neurodevelopmental Diagnostic Center for Young Children in Davidsonville.	22 boys 49, XXXXY with or without hormonal treatment.	The mean age of treatment for: Group 1 was 12 months with the mean age of first evaluation 74 months. The mean age of first evaluation for Group 2 was 87 months.	Leiter International Performance Scale-Revised (LIPS-R), Preschool Language Scales-4 (PLS-4), the Receptive One Word Vocabulary Test (ROW- PVT-R), Expressive One Word Picture Vocabulary Test- Revised (EOWPVT-R), MacArthur Communication Developmental Inventory (CDI), Gilliam Autism Rating Scale-2 (GARS-2).	Significant positive treatment effect in speech and language domain, gestural communication and vocabulary development. No treatment effect was seen in nonverbal capacities.
**Lee et al., 2012 [2]**	Patients recruited through advertisements via the NIH website and parent-support groups across North America. Controls were recruited from the US in a brain development study.	110 youth with X/Y-aneuploidies (32 female) and 52 with typical development (25 female)	Mean age ~12 years	Wechsler Abbreviated Scale of Intelligence (WASI), Wechsler Preschool and Primary Scale of Intelligence–Third Edition (WPPSI-III), Children’s Communication Checklist-2, Social Responsiveness Scale.	Both supernumerary X- and Y-chromosomes were related to depressed structural and pragmatic language abilities and increased autistic traits. An additional Y-chromosome had a greater impact on pragmatic language; the addition of one or more X-chromosomes had a greater impact on structural language.
**Cordeiro et al., 2012 [12]**	Children’s Hospital Colorado (Denver), Thomas Jefferson University-Philadelphia (TJU).	102 Males with XXY 40 XYY 32 XXYY	XXY: Mean age 10.8 years XYY: Mean age 9.93 years XXYY: Mean age 11.57 years	(1) Social Responsiveness Scale: Parent Report Questionnaire (SRS-P); (2) Cognitive testing: TJU—Differential Ability Scales–2nd edition (DAS-2). Denver—Wechsler Abbreviated Scale of Intelligence(WASI); Wechsler Intelligence Scale for Children–4th Edition (WISC-IV).	An additional Y chromosome may contribute to increased risk of autistic behaviours.
**Tartaglia et al., 2012 [8]**	Patients seen from 2004 to 2010 at the University of California—Davis MIND Institute and at Children’s Hospital Colorado.	167 participants (XXY *n* = 56, XYY *n* = 33, XXX *n* = 25, XXYY *n* = 53)	Age 6–20 years	(1) Cognitive levels: Wechsler Abbreviated Scale of Intelligence (WASI), Wechsler Intelligence Scale for Children–Third Edition (WISC III), Wechsler Intelligence Scale for Children– Fourth Edition (WISC IV), Wechsler Adult Intelligence Scale–Third Edition (WAIS III); (2) Adaptive functioning: Vineland Adaptive Behaviour Scales-II, Adaptive Behaviour Assessment System—Second Edition (ABAS-II), Scales of Independent Behaviour—Revised (SIB-R); (3) ADHD: Conners’ Rating Scale, Swanson, Nolan, and Pelham Questionnaire—Fourth Edition (SNAP-IV).	58% met DSM-IV criteria for ADHD on parent-report questionnaires. The Inattentive subtype was most common in XXY and XXX, whereas the XYY and XXYY groups were more likely to also have hyperactive/impulsive symptoms. Psychopharmacologic treatment with stimulants was effective in 78.6% (66/84).
**Blumenthal et al., 2013 [13]**	Patients recruited with the help of Neurodevelopmental Diagnostic Center for Young Children and a parent advocacy group; controls recruited from the community through the National Institutes of Health (NIH) Normal Volunteer Office, newspaper advertisements, and outreach to schools in the Washington, DC area.	14 patients with 49, XXXXY 42 healthy controls	Mean age: 11.6 years	Wechsler Abbreviated Scale of Intelligence (WASI), Peabody Picture Vocabulary Test-4 (PPVT-4), Physical And Neurological Examination for Soft Signs (PANESS), Adaptive Behaviour Assessment System (ABAS), Child Behaviour Checklist (CBCL), MRI.	Increased dosage of genes on the X chromosome has adverse effects on white matter development.
**Tartaglia et al., 2017 [14]**	Participants from two sites (University of California (UC) Davis MIND Institute in Sacramento, California and Thomas Jefferson University (TJU) in Philadelphia, Pennsylvania)	XXY/KS (*n* = 20) XYY (*n* = 57) XXYY (*n* = 21)	Age 3–25 years.	(1) Cognitive levels—UC: Mullen Scales of Early Learning (MSEL), Wechsler Abbreviated Scale of Intelligence (WASI); TJU: Differential Abilities Scale-2nd Edition (DAS-2) (2) Adaptive functioning: Vineland Adaptive Behaviour Scales, 2nd Edition Interview Edition; (3) ASD: the Social Communication Questionnaire (SCQ), the Social Responsiveness Scale (SRS), and the Autism Diagnostic Observation Schedule (ADOS). the Autism Diagnostic Interview-Revised (ADI-R).	Males with Y chromosome aneuploidy (XYY and XXYY) were 4.8 times more likely to have a diagnosis of ASD than the XXY/KS group, and 20 times more likely than males in the general population.
**Udhnani et al., 2018 [9]**	Patients included in the Research program conducted at the National Institute of Mental Health (NIMH) Intramural Research Program.	79 youth with SCAs and 42 typically developing controls	Age 6.60–22.60 years	Phonemic and semantic fluency conditions, Wechsler Abbreviated Scale of Intelligence, Wechsler Preschool and Primary Scale of Intelligence—Third edition.	Both supernumerary X and Y chromosomes were associated with verbal fluency deficits relative to controls. These impairments increased as a function of the number of extra X chromosomes. Whereas one supernumerary Y chromosome was associated with similar performance across fluency conditions, one supernumerary X chromosome was associated with relatively stronger semantic than phonemic fluency skills.
**Burgemeister et al., 2018 [15]**	6/8 patients were participants of the 2015 and 2017 annual meetings of families with boys and men with 49,XXXXY syndrome; 1/8 was evaluated at the genetikum genetic practice in Stuttgart and one patient was seen and evaluated at the Institute of Human Genetics in Berlin.	8 boys and men with 49, XXXXY	Age 3–24 years	HAWIK IV (Hamburg Wechsler Intelligence Test for Children, Version IV), SON-R (Snijders-Oomen Nonverbal Intelligence Test), K-ABC (Kaufmann Assessment Battery for Children), neurological and physical examination	Increasingly perceptible distinct facial gestalt over time; muscular hypotonia, radioulnar synostosis, white matter anomalies, fifth-finger clinodactyly, recurrent respiratory infections in early childhood and teeth anomalies. IQ scores ranged between 40–70.
**Srinivasan et al., 2019 [16]**	IMAGINE ID project	15 children with XXYY and 30 controls;	4–14 years	(1) Development and Well-being Assessment; (2) Strengths and Difficulties Questionnaire; (3)Everyday Feelings Questionnaire	Children with XXYY experienced significantly more frequent and intense temper outbursts than the control group.
**Gropman et al., 2020 [17]**	Patients received clinical evaluation as part of an annual conference for children diagnosed with 49,XXXXY and their families.	67 boys with 49, XXXXY	Age 5 months–10 years	The Mental Development Index (MDI)/Cognitive domain of the Bayley Scales of Infant and Toddler Development (BSID), Second or Third Edition; Leiter International Performance Scale (LIPS), Revised or Third Edition; Wechsler Scales of Intelligence for Verbal IQ.	Higher neurocognitive capacities, both verbally and non-verbally, than previously reported. Infant boys who received early hormonal therapy had significantly higher scores on the cognitive domain of the Bayley Scales of Infant Development.
**Samango-Sprouse et al., 2020 [18]**	Patients evaluated by The Focus Foundation in collaboration with the Neurodevelopmental Diagnostic Center for Children; not every child was in attendance each year.	72 boys with 49, XXXXY	Preschool/School-aged/Adolescence	Beery Buktenica Developmental Test of Visual-Motor Integration, Sixth Edition, the Bayley Scales of Infant and Toddler Development, Third Edition (BSID-III), and the Bruininks–Oseretsky Test of Motor Proficiency, Second Edition.	Truncal and extremity hypotonia significantly impact on motor milestones and ambulation; dysmorphic features include epicanthal folds, frontal bossing and synophrys. Visual perception skills are mildly impaired and cranial nerves are typically intact. Preschool boys treated with testosterone replacement had significantly increased scores on the BSID-III Psychomotor Development Index.
**Lasutschinkow et al., 2020 [19]**	Patients received clinical evaluation by The Focus Foundation in collaboration with the Neurodevelopmental Diagnostic Center for Children from 2004 until 2018.	69 boys 49, XXXXY	Preschool/School-aged/Adolescence	Child Behaviour Checklist (CBCL), Social Responsiveness Scale, second edition (SRS-2), Behaviour Rating Inventory for Executive Function, second edition (BRIEF-2), Behaviour Rating Inventory for Executive Function-Preschool Edition (BRIEF-P).	Deficits in social cognition and communication beginning in preschool, with consistent social awareness and motivation for social activities; signs of anxiety presented during preschool years and increased in severity with age, particularly in internalizing problems.
**Thompson et al., 2020 [1]**	This study was part of an international survey to assess early therapies, school supports, and educational outcomes for children with SCA conditions. Participants were recruited through social media websites and email lists for the eXtraordinarY Kids Clinic of Children’s Hospital Colorado and the Association for X and Y Chromosome Variations (AXYS). Data were collected between 15 May and 15 July 2019.	*n* = 105	Children not yet entered into Kindergarten	Closed-ended and multiple-choice questions to describe the supports for clinical population with demographic questions. Open-ended questions were also included to suggest questions about parent perspectives on supports for young children with SCAs.	Public early childhood intervention services with speech therapy as the most common service. Parents described interventions as desirable and effective yet also difficult to obtain due to issues with the SCA phenotype, lack of provider knowledge, and challenges navigating the intervention systems.
**Samango-Sprouse et al., 2021 [20]**	Patients eeceived clinical evaluation over the last 16 years as part of an annual conference for children and families hosted by the Focus Foundation in collaboration with the Neurodevelopmental Diagnostic Center for Children.	85 boys with 49, XXXXY	Age 3 months–10.7 years	The Bayley Scales of Infant and Toddler Development; The Preschool Language Scales (PLS); The Expressive One-Word Picture Vocabulary Test (EOWPVT); Receptive One-Word Picture Vocabulary Test (ROWPVT); The Early Language Milestones Scale-2 (ELM-2).	Increased incidence (91.8%) of Childhood Apraxia of Speech. Differences were demonstrated between boys who received early hormonal treatment and untreated boys on the language scales of the Bayley Scales of Infants and Toddlers and on the Expressive One Word Picture Vocabulary Test.
**Lee et al., 2021 [21]**	Patients referred by their physicians for comprehensive neurodevelopmental evaluations.	29: 47, XXY, 27: 49, XXXXY, 14: 46, XY	XXY: Mean age 12 years XXXXY: Mean age 8.6 years XY: no data	Behaviour Rating Inventory of Executive Function (BRIEF) and Child Behaviour Checklist (CBCL). Assays to compare saliva DNA between the 49,XXXXY and neurotypical 46, XY groups for methylation diferences in a small selection of genes on the X chromosome.	Higher levels of CpG methylation at regulatory intronic regions in X-linked genes encoding the androgen receptor (AR) and monoamine oxidase A (MAOA) may be linked to (externalizing) behaviour in boys with 49, XXXXY.

## Data Availability

Not applicable.

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
