# Peer review of "Clinical, Cognitive and Neurodevelopmental Profile in Tetrasomies and Pentasomies: A Systematic Review"

_children, 2022, doi:10.3390/children9111719_

Round 1
Reviewer 1 Report
Review comment
Addressing a specific issue in the study was the strength of the research. I think that examining the articles investigating a syndrome with a low prevalence will contribute to the readers and the literature.
Although the subject selection is good, there are problems in the writing and presentation of the manuscript. Manuscript is not well-suited to a systematic review template. In this sense, it is not possible for the reader to draw a conclusion from the compilation under these conditions. Therefore, a number of changes are required.
Major revision
1. The abstract part contained only the general information explanation in the introduction part of an article and the purpose of the study. It is not correct to write a summary in this way. The summary should contain information about the method, findings, and conclusions. When the reader looks at the abstract, he or she should have brief information about all parts of the article. For this reason, the summary should be rearranged by completing these missing parts.
2. The systematic compilation should be written in a specific template. However, when we looked at the manuscript, especially the summary, method and conclusion parts were very inadequate. It is very important for a systematic review to present the articles in the literature properly so that the reader can understand the review. A checklist such as PRISMA is used to present the systematic compilation in a standardized manner and in a specific template. In this sense, the compilation should be rearranged using a checklist in order to bring the presentation to a higher quality.
3. Very little information was given in the method section. It should be written in much more detail. While the feature of the language of the articles should be given here, it is given in the conclusion part. Each filtering used in the selection of the article should be given in the method section.
4. It is just as important to present the reviews in a systematic way, by examining the quality of the evidence of the articles included in the review. In this sense, scales such as the PEDro scale and the Newcastle Ottawa Scale and the quality levels of the articles should be evaluated and presented in the review.
5. How many people did the article search and was it done by different people at the same time? The fact that the search is carried out by different researchers is important for another researcher to notice the article that one author missed during the search. Information on this topic should be added to the article.
6. It is very important that the results are presented in a beautiful way. However, when the manuscript was examined, this part was written with very insufficient brief information. There is almost no information about the articles you included in the compilation. The year of these articles, in which country they were made, with how many people, what subject they contain, what method the evaluation was made, what results were found, and the individuals in which age range, etc. should be stated in tables.
7. By categorizing the articles you have chosen, you should tabulate which one is neurocognitive, behavioral or linguistic in the results section. In the discussion, you should discuss these articles with their data in this way.
8. The discussion section should be written according to the information of the articles given in the results. When we look at the manuscript, the discussion was written without any information about the articles in the results. Articles should be tabulated and added to the results section, and the discussion should be rearranged in line with these results.

Author Response
Dear Editors,
Thank you for your comments and suggestions.
Here the arrangements we provided to the article, following the reviewers’ comments:
Major revision
- We rearranged the abstract of the article explaining the methods of the study, results and conclusions
- We rearranged the compilation using checklist PRISMA.
- We revised method section, writing much more details and explaining each filtering used in the selection of the article in the method section.
- We focused our interest on an extremely rare condition with very little standardization. Thus it was impossible and counterproductive for us to examine the quality of the evidence of the articles included. We tried our best to improve the quality of the review not including single case report or articles with less than 3 patients.
- The literature search was performed by two authors independently, as reported in the Materials and Method section 2 (2.1. Literature search)
- We added tables to summarize informations about the articles included in the review in order to present results in a more systematic way, A new analysis of the literature used for the article was made to recollect every important information related to single studies to better evaluate background, procedures and conclusions of these works. The aim was to create a more comprehensible list of data, which could be useful to understand results and conclusions of the present paper.
7, 8. By categorizing the articles we have chosen, we tabulate which one is neurocognitive, behavioral or linguistic in the results section (Table 1). In the discussion, we summarized each main topic of selected articles in specific paragraph (psychosocial and behavioral aspects, cognitive profile, neuroimaging studies etc.) to give a wholesome vision of characterization of these conditions.
Reviewer 2 Report
The aim of Clinical, Cognitive and Neurodevelopmental Profile In Tetrasomies and Pentasomies: A Systematic Review was to discuss the collection of literature regarding the developmental abnormalities in patients with supernumerary X or Y chromosomes. The authors chose to focus on tetrasomies, like 48,XXYY, and pentasomies, like 49,XXXXY.
Additionally,
- Several statements, especially regarding the results from the literature, are not cited.
- In conjunction with several inaccuracies and misinformation, it is not appropriately organized or well-written. This paper fails to contribute to the field of rare chromosomal disorders because of these faults.
- There is an excessive amount of additional citations needed, grammatical and spelling errors, and exclusion of literature that refutes their hypothesis.
Abstract
- Throughout the article, the word “behavioral” is spelled different ways (i.e. behavioural). To promote consistency, the authors should ensure they are using the same language.
- On line 19, the sentence “The addition of extra X…psychological disorders,” is almost copied word-for-word from line 50 in the Introduction. The authors should exercise more variability in their word choice in order to not plagiarize themselves.
Introduction
- On line 43, the authors state “49,XXXXY is the rarest form, with fewer than 10 cases reported to date.” The location and time of this data collection should be specified because there are more than 100 cases currently reported. This statement is misinformed and reduces the integrity of the authors.
- On line 57, “with an increase…fluency impairments,” should have an accompanying citation.
- On line 65, “Decreased motor control…and 49,XXXXY,” should have an accompanying citation.
- On line 66, “Due to motor planning deficits…single words and phrases,” should have an accompanying citation.
- On line 68, “Language formulation…receptive skills,” should have an accompanying citation.
- On line 69, “The severity of language-based…and oppositional behavior,” should have an accompanying citation.
Materials and Methods
- From line 84 to 87, the variability of spacing between many of the words is distracting to the reader and should be fixed.
- On line 97 to 98, it mentions that Figure 1 includes the selection process of their 23 articles. In the figure itself, it concludes 25 articles were selected. The authors should fix this discrepancy.
Results
- Other than Figure 1, there are no other figures or tables organizing the data from the literature search. The authors should include this to make the conclusions of the paper more prominent.
- In addition to the lack of figures and tables, no specific scores were included in the Results section. For example, the authors talked about discrepancies between expressive and receptive scores, but did not provide specific numbers from the collected literature to support their statements.
Discussion
- While I think the organization of separating the different disorders is a good idea, a majority of the information in the Discussion section should be in the Results section instead. The Discussion section should not be reiterations of the data collected from the literature, but rather the author’s conclusions of this data. This section should be completely redone. An example is line 245, “There was a disproportionate excess of both grey matter and white matter in the parietal lobe.”
- On line 194, “Furthermore, even patients…seen in ASD,” should have an accompanying citation.
- On line 202, “ADHD symptoms…the most common,” should have an accompanying citation.
- On line 205, “Attentional difficulties…testosterone deficiency,” should have an accompanying citation.
- On line 208, “More than 70% of children…to treatment discontinuation,” should have an accompanying citation.
- On line 223, “Some boys who…the low normal range,” should have an accompanying citation.
- On line 227, “Boys with 48,XXYY…to boys with 48,XXXY or 49,XXXXY,” should have an accompanying citation.
- On line 232, “In a cohort of patients…cognitive (IQ) scores alone,” should have an accompanying citation.
- On line 316, “Contrasting personality…temper tantrums,” should have an accompanying citation.
- On line 327, “In addition, signs of anxiety…elevated scores,” should have an accompanying citation.
- On line 352, “Delayed neuromotor…verbal motor functions,” should have an accompanying citation.
Author Response
Dear Editors,
Thank you for your comments and suggestions.
Here the arrangements we provided to the article, following the reviewers’ comments in report 2:
- We have replaced all the words “bahavior” and “behavioral” with “behaviour” and “behavioural”.
- We have differentiated the sentences on line 19 and 50 (in the Introduction).
- We modified the sentences where suggested.
- We corrected the variability of spaces from line 84 to 87.
- We have corrected the previous discrepancy modifying the number of articles resulting from the selection.
- We inserted a summarizing table to report the characteristics of selected articles.
- We organized the Discussion section of our review reporting all data deriving from literature in different paragraphs, to lead the readers to a better knowledge of the topic.
- We reported all citations required for all indicated sentences.
Round 2
Reviewer 1 Report
Reviewer Comments
Dear researcher;
Although you can see the revisions that have been made from the revisions I have previously suggested, it is seen that there are incomplete and unmade revisions. I must say that in your manuscript that I received, the English corrections are also in the text. I received a very complex and confusing text, you needed to submit a manuscript in a much more careful and orderly manner. The flowchart is not readable, the tables are mixed and the result part is not complete. I am waiting for you to submit a manuscript in a more systematic review format by making these and similar arrangements again. Your manuscript is not sufficient to be published as it is. Please make the revisions I have given more carefully.
Revision
Page 1, line 33-34; “The aim of this review was to provide a comprehensive view of the neurocognitive, behavioural, and social aspects of these syndromes.” This purpose sentence that you wrote in the result part of the summary section should be in the purpose section of the summary section. Since you stated this in the purpose part of the summary, delete it here, there is no need to rewrite it.
1. Page 2, line 63-64; “Development” These keywords have a very general meaning. The keyword you use must be target oriented. Either delete it or use it as a developmental disorder, neurodevelopmental disorder. Because you mentioned these disorders throughout the manuscript. Keywords should be related to the topics you highlight.
2. Page 2, line 64-75; Sources 4 and 5 in the introduction are very old. A source where prevalence information is given should be a journal. Modify this resource.
3. The quality of the articles included in the study must be evaluated. You have decided that it is impossible to evaluate the quality of the article. The quality of each article can be measured. We cannot interpret this as a situation that will weaken the study. In order for future studies to be more qualified, studies in the literature should be presented in detail. Although the quality of the included articles is inadequate, this should be noted as a limitation.
4. “Figure 1” The flowchart (figure 1) should be the flowchart of PRISMA. Change the flowchart. This is important for standardization. The text of this figure is not readable. It should be arranged in a more legible way.
5. The information of the articles given in the tables in the result section is very mixed and there is no order. The average age of some participants is given, and some are not. A number of adjustments should be made for a better follow-up understanding of the articles. You do not need to write the name of the center as the place where the research was conducted. It is sufficient to write in which city the research was conducted and should be stated in a separate column. The number of participants and their age range should also be shown in separate columns within the table. Thus, the article data can be read more clearly. The presentation of all headings in the table should be standardized.
6. Sort the articles you mentioned in the table in the result section according to the year. The data must have a certain form of presentation. Sort by year from oldest to newest.
7. In the result part, no explanation is given for the table containing the article information. There was no table in your previous manuscript, no text in your current manuscript. If a table is given in the result part of an article, a not too long general explanation of the table should also be given. It should not be presented as such without any explanation. Information containing table information should be added to the result part (table 1).

Author Response
Dear Editor,
we revised the flowchart, the tables, and the result part in a more systematic way, as suggested:
- Page 1, line 33-34; We put the sentence “The aim of this review was to provide a comprehensive view of the neurocognitive, behavioural, and social aspects of these syndromes.” in the purpose section of the summary section and deleted it from the lines indicated.
- Page 2, line 63-64; We used “developmental disorder” instead of “development” as keyword, to be more target oriented and to related it to the topics highlighted n the article, as suggested.
- Page 2, line 64-75; Sources 4 and 5 in the introduction are very old. A source where prevalence information is given should be a journal. Modify this resource.
- We added informations about the quality of the included articles.
- “Figure 1” we changed the flowchart for a better standardization.
- We adjusted the information of the articles given in the tables in the result section for a better follow-up understanding of the articles and standardized the presentation of all headings in the table.
- We sorted the articles you mentioned in the table in the result section according to the year..
- We modified the result part as requested.
Reviewer 2 Report
This paper has shown much improvement through the revision process. My only suggestion is the abstract should include another sentence or two about HRT. There are strong studies that have shown the benefit of HRT so we suggest more be added than just "HRT is usually recommended for these patients." Additional studies on the impact of HRT in these disorders and how Testosterone replacement positively affects these kids should be added.
Author Response
Dear Editor,
thank you for your suggestion to improve our work.
As suggested, we added more information in the abstract about HRT.